# Generation and Comprehension Hand-in-Hand: Vision-guided Expression Diffusion for Boosting Referring Expression Generation and Comprehension

**Jingcheng Ke**[1]    **Jun-Cheng Chen**[2]    **I-Hong Jhuo**[3]    **Chia-Wen Lin**[1]    **Yen-Yu Lin**[4]
[1]National Tsing Hua University    [2]Academia Sinica
[3]Microsoft AI    [4]National Yang Ming Chiao Tung University

## Abstract

Referring expression generation (REG) and comprehension (REC) are vital and complementary in joint visual and textual reasoning. Existing REC datasets typically contain insufficient image-expression pairs for training, hindering the generalization of REC models to unseen referring expressions. Moreover, REG methods frequently struggle to bridge the visual and textual domains due to the limited capacity, leading to low-quality and restricted diversity in expression generation. To address these issues, we propose a novel **VI**sion-guided **E**xpression **D**iffusion **M**odel (VIE-DM) for the REG task, where diverse synonymous expressions adhering to both image and text contexts of the target object are generated to augment REC datasets. VIE-DM consists of a vision-text condition (VTC) module and a transformer decoder. Our VTC and token selection design effectively addresses the feature discrepancy problem prevalent in existing REG methods. This enables us to generate high-quality, diverse synonymous expressions that can serve as augmented data for REC model learning. Extensive experiments on five datasets demonstrate the high quality and large diversity of our generated expressions. Furthermore, the augmented image-expression pairs consistently enhance the performance of existing REC models, achieving state-of-the-art results. The source code is available at `https://github.com/freedom6927/VIE-DM.git`.

## 1 Introduction

Joint vision-language representation learning has become a burgeoning research focus in computer vision, with diverse applications such as referring expression comprehension (REC) Su et al. (2023); Yang et al. (2023); Su et al. (2024), referring expression generation (REG) Sun et al. (2023), referring expression segmentation Sun et al. (2023); Liang et al. (2022), visual question answering Dancette et al. (2023), image captioning Luo et al. (2023), and scene understanding Peng et al. (2023). Among these applications, REG and REC are essential and interconnected components of joint visual-textual reasoning, enabling the identification of target objects specified by given expressions.

REG aims to generate a precise textual description of a specific object within an image, while REC identifies the corresponding object in the image based on a given expression. However, REC models frequently encounter difficulties with novel referring expressions due to the scarcity of image-expression training pairs. To mitigate this issue, we explore its complementary REG task by introducing a diffusion-based method that generates multiple diverse referring expressions for a target object. These generated expressions can be employed to augment training data for REC models. Meanwhile, existing REG methods such as Mao et al. (2016); Yu et al. (2016; 2017); Liu et al. (2017; 2020); Tanaka et al. (2019); Niu et al. (2021); Sun et al. (2023) are developed based on transformer-LSTM or CNN-LSTM frameworks. These methods directly input visual features from transformers or CNNs into the LSTM for expression generation. However, they do not effectively bridge the gap between the visual and textual domains. In addition, existing REG methods are deterministic and fail to generate diverse expressions needed to fulfill our goal, expression augmentation.

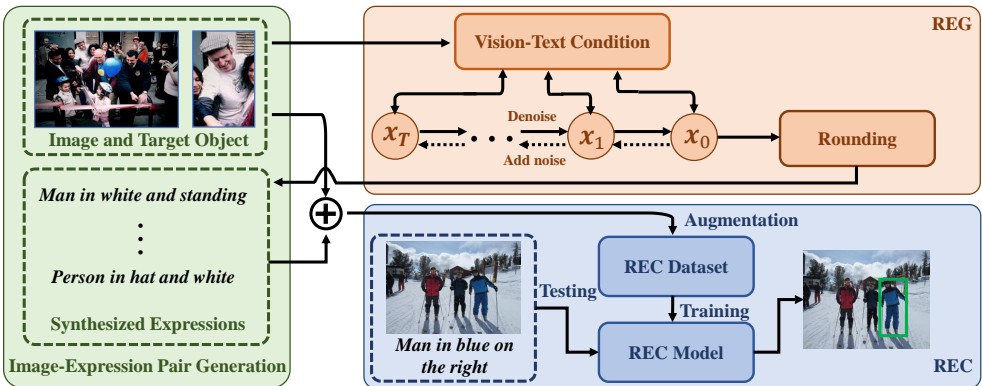

Figure 1: **Our vision-guided expression diffusion model for referring expression generation.** The proposed vision-text condition (VTC) module takes an image and the target object as input. It integrates these visual conditions into the diffusion process using cross-attention at each denoising step, and can generate diverse and high-quality referring expressions that tightly align with the visual context of the target object. The generated image-text pairs are then used to augment the training data to improve the REC performance.

The diversity of synthesized samples is a vital attribute for a generative model Song et al. (2021); Ho et al. (2020), as it determines the spectrum of possibilities the model can encompass and predict. A model exhibiting high diversity can generate a wide variety of samples, effectively capturing the full spectrum of features in the data. In the past, certain machine translation methods, such as back translation Gu et al. (2018); Saharia et al. (2020), using non-autoregressive language models (LMs) have shown the ability to produce diverse sentences. However, these methods are tailored for speech and translation contexts, where the entropy among valid outputs is typically low. It has been demonstrated that these approaches are ineffective for language modeling Ren et al. (2020). Recently, text diffusion models Gong et al. (2023); He et al. (2023) have gained widespread adoption in sentence generation tasks owing to their interpretability and substantial diversity. These models are used in various applications, such as text simplification, question generation, and paraphrasing.

As depicted in Figure 1, building on the success of text diffusion models and addressing the aforementioned REG challenges, we introduce a novel method, VIsion-guided Expression Diffusion Model (VIE-DM), for REG. The diffusion model in VIE-DM incorporates a vision-text conditioning (VTC) module and a transformer decoder for conditional text generation. The VTC module implements two key strategies: cross-attention and token selection. In each denoising step, the cross-attention strategy is employed to infuse the visual features of the image and target object into the noisy text features of the expression. Subsequently, the token selection strategy is applied to concentrate on important features of the image. After processing via the VTC module, the output is fed into the transformer decoder to predict the noise and generate denoised features. This iterative process continues until the specified number of denoising steps is reached.

The VTC module facilitates enhanced alignment between the noisy text features and the target object's context compared to conventional CNN-LSTM or Transformer-LSTM frameworks, which are constrained by their model capacity. Additionally, this advantage enables the proposed method to generate more precise referring expressions for the target object in the REG task, outperforming random expressions produced by unconditional text diffusion models. Guided by the visual features of the image and the target object, we can generate diverse referring expressions adhering to the target object by randomly sampling a group of Gaussian noise vectors followed by iteratively denoising steps using our model. The resulting image-expression pairs are subsequently used to augment the training set and enhance REC model training.

This paper makes two primary contributions. First, to the best of our knowledge, we are the first to introduce the diffusion model to the REG task. By incorporating our vision-text condition module, the visual features of the image and target object are more effectively aligned with the noisy text throughout the forward and backward diffusion processes. This enables the model to generate more accurate and diverse expressions for the target object in the image. Second, we substantiate our

claim with a comprehensive evaluation across five datasets, demonstrating that our diffusion model can effectively augment the REC datasets to consistently improve different REC methods.

## 2 RELATED WORK

**Referring Expression Generation (REG).** REG and image captioning share some similarities. REG generates a description for a target object, focusing on a particular region of an image, whereas image captioning produces expressions that describe the entire image. For REG, Mao et al. (2016) integrate deep learning into REG. They extract visual features and enable expression generation using an LSTM. Additionally, they introduce a Maximum Mutual Information (MMI) training strategy, which has become a standard approach in many REG models. To better bridge the visual and textual domains, some studies include additional annotations that detail the visual and positional differences among objects. For instance, Liu et al. (2017; 2020) use attribute labels to establish connections between the objects in an image and the corresponding expressions before generation.

Recently, some methods Yu et al. (2017); Sun et al. (2023) consider REC and REG jointly. Yu et al. (2017) introduce a Listener-Speaker system, where the Speaker module generates expressions guided by an REC model, while the Listener module identifies the object in the image based on the provided expression. Sun et al. (2023) utilize two types of cross-attention modules to connect REC and REG: One combines the image and expression (REC), while the other integrates the image and target object (REG). Subsequently, the outputs of the cross-attention modules are fed into a fusion module to determine the target object's location and generate synonymous expressions.

Existing REG methods commonly employ CNN-LSTM or transformer-LSTM frameworks for expression generation. However, they often struggle to capture complex attributes within expressions due to the deterministic nature and limited model capacity of LSTM, thereby limiting data diversity. While existing REG methods employ the Maximum Mutual Information (MMI) training strategy to distinguish the target object from other irrelevant objects in the image, MMI is only suitable for deterministic methods and cannot be directly applied to diffusion models. In contrast, our method introduces an image-guided text-diffusion model for REG. Leveraging the stochastic advantage of diffusion models, our approach significantly enhances the quality and diversity of generated expressions. Furthermore, we introduce a token selection strategy that helps our method better capture essential features, thereby reducing the impact of irrelevant features in the image.

**Diffusion Models for Sentence Generation.** Several studies have explored conditional diffusion models for sentence generation. For instance, Diffusion-LM Li et al. (2022) is the pioneering work in text diffusion modeling, focused on developing language models with complex and fine-grained behavior controls. Unlike Diffusion-LM, DiffusER Reid et al. (2023) integrates Levenshtein operations into its diffusion model and generates more linguistically diverse sentences while preserving the original semantics. Lin et al. (2023) further introduce a large-scale pre-trained diffusion language model comprising an encoder and a diffusion-based decoder. Zheng et al. (2023) propose a reparameterization strategy that enhances the efficiency of discrete diffusion probabilistic models in natural language generation. Gong et al. (2023) propose an end-to-end classifier-free conditional diffusion model that combines the Seq2Seq framework with the diffusion model for sentence generation. He et al. (2023) present DiffusionBERT, which investigates the training of BERT to understand the reverse process of the diffusion model within the discrete domain.

However, existing conditional text diffusion models only use additional text descriptions to generate sentences and do not consider using images as a condition for the REG task. In this paper, we introduce VIE-DM for the REG task. Guided by both the image and the target object, the proposed VIE-DM generates high-quality and diverse synonymous expressions. To the best of our knowledge, we make the first attempt to implement conditional text diffusion models for the REG task.

**Transformers for Referring Expression Comprehension (REC).** Recent state-of-the-art REC algorithms predominantly rely on transformers. Existing REC models often struggle to generalize to unseen referring expressions, as they are trained on fixed image-expression pairs. Although SelfEQ He et al. (2024) attempts to paraphrase the expressions by synonym replacement for augmentation, it exhibits limitations in the diversity and complexity of the generated image-expression pairs for REC, as acknowledged in its paper. To address this issue and further improve the performance of REC methods, we augment the REC training set with the image-expression pairs generated

Figure 2: **Training and sampling phases of our proposed VIE-DM**. During training, noisy text tokens of the expression and the visual tokens of the image and the target object serve as the input to the VTC (vision-text condition) module. The VTC module compiles noisy vision-text features, which are passed to a transformer decoder for noise prediction at each denoising step. During sampling, we generate a new expression by sampling a random noise conditioned on the visual features of the given image and the target object through VTC followed by the transformer decoder for denoising. The noisy vision-text features undergo iterative denoising. Ultimately, we use the rounding operation Reid et al. (2023) to decode the denoised features and yield a new expression.

by our method. Transformer-based REC methods are then applied to the augmented dataset during both training and testing.

## 3 METHOD

Figure 2 illustrates our REG framework, detailing both the training and sampling phases. Our proposed VIE-DM denoising network mainly consists of (1) a vision-text condition (VTC) module to condition the noisy text features on the embedded visual features of the entire image and the patches of the target object and (2) a transformer decoder to take the interacted feature for noise prediction at each denoising step. Our VIE-DM is elaborated as follows.

### 3.1 DIFFUSION MODELS

Diffusion models are latent variable models, involving a forward and a reverse diffusion process. Given a sample $x_0 \sim q(x_0)$, the forward process entails producing a Markov chain of latent variables $x_1, ..., x_T$ by iteratively introducing small increments of Gaussian noise to the sample:

$$q(x_t|x_{t-1}) = \mathcal{N}\left(x_t; \sqrt{1-\beta_t}x_{t-1}, \beta_t \mathbf{I}\right), \tag{1}$$

where $\mathbf{I}$ is an identity matrix, and $\{\beta_t \in (0, 1)\}_{t=1}^T$ represents the variance schedule that governs the step size for introducing noise. Following $T$ steps, $x_T$ converges to an isotropic Gaussian distribution. If $\beta_t$ is sufficiently small, the reverse process $q(x_{t-1}|x_t)$ can be approximated as a Gaussian distribution, which can be learned using a parametric model:

$$p_\theta(x_{t-1}|x_t) = \mathcal{N}(x_{t-1}; \mu_\theta(x_t, t), \Sigma_\theta(x_t, t)), \tag{2}$$

where $\mu_\theta$ and $\Sigma_\theta$ are trainable parameterized models. When conditioning on $x_0$, $q(x_{t-1}|x_t, x_0)$ has a closed-form solution, the corresponding objective can be simplified to

$$\mathcal{L}_{\text{DM}} = \mathbb{E}_{x_0, \varepsilon \sim \mathcal{N}(0, \mathbf{I}), t}\left[||\varepsilon - \varepsilon_\theta\left(\sqrt{\bar{\alpha}_t}x_0 + \sqrt{1-\bar{\alpha}_t}\varepsilon, t\right)||_2^2\right], \tag{3}$$

where $t \in \{1, ..., T\}$ denotes the time index of each denoising step, $\varepsilon \sim \mathcal{N}(0, \mathbf{I})$ is Gaussian noise, $\varepsilon_\theta$ is the function for noise prediction from $x_t$, $\alpha_t = 1 - \beta_t$, and $\bar{\alpha}_t = \prod_{i=1}^t \alpha_i$.

## 3.2 PRELIMINARY

Given an image of size $H \times W \times 3$, an image patch of the target object of size $H_o \times W_o \times 3$, and an expression comprising $L$ words, we feed the entire image and the target object into a vision transformer, resulting in two groups of visual tokens: $\mathbf{v} = \{v_i\}_{i=1}^{N}$, $v_i \in \mathbb{R}^d$ for the image and $\mathbf{o} = \{o_i\}_{i=1}^{N}$, $o_i \in \mathbb{R}^d$ for the target object, respectively. The expression undergoes processing in BERT to yield $L$ text tokens representing the words, denoted by $\mathbf{e} = \{e_i\}_{i=1}^{L}$. In addition, we employ an embedding function $E_{\mathrm{MB}}$ and normal distribution to transform the discrete representations $\mathbf{e}$ into continuous ones $\mathbf{z}_0$, i.e., $\mathbf{z}_0 = E_{\mathrm{MB}}(\mathbf{e})$, by following Li et al. Li et al. (2022).

## 3.3 FORWARD AND REVERSE PROCESSES FOR EXPRESSION DIFFUSION

**Forward Process.** Let the visual guidance from the image and the target object be the condition $\mathbf{c} = [\mathbf{o}, \mathbf{v}]$. At the $t$-th forward step, the noisy text feature $\mathbf{z}_t$ is generated by adding noise to $\mathbf{z}_{t-1}$ through $q(\mathbf{z}_t|\mathbf{z}_{t-1}, \mathbf{c}) = \mathcal{N}(\mathbf{z}_t|\mathbf{c}; \sqrt{1-\beta_t}\mathbf{z}_{t-1}|\mathbf{c}, \beta_t\mathbf{I})$, where $\{\beta_t\}_{t=1}^{T}$ are the noise schedule.

**Reverse Process.** The reverse process involves recovering the original $\mathbf{z}_0$ by iteratively denoising the noisy text features from $\mathbf{z}_T \sim \mathcal{N}(0, \mathbf{I})$ under the guidance of the entire image and target object. The reverse process is also defined as a Markov chain and its joint distribution is formulated as $p_\theta(\mathbf{z}_{0:T}|\mathbf{c}) := p(\mathbf{z}_T)\prod_{t=1}^{T} p_\theta(\mathbf{z}_{t-1}|\mathbf{z}_t, \mathbf{c})$. $\mathbf{z}_{t-1}$ is sampled according to $p_\theta(\mathbf{z}_{t-1}|\mathbf{z}_t, \mathbf{c}) = \mathcal{N}(\mathbf{z}_{t-1}; \mu_\theta(\mathbf{z}_t, \mathbf{c}, t), \Sigma_\theta(\mathbf{z}_t, \mathbf{c}, t))$, where $\mu_\theta$ and $\Sigma_\theta$ parameterize the predicted mean and variance of $p_\theta(\mathbf{z}_{t-1}|\mathbf{z}_t, \mathbf{c})$, respectively. It can be further reduced to noise prediction similar to what mentioned earlier and modeled by our denoising network consisting of a vision-text condition (VTC) module and a transformer decoder, $\varepsilon_\theta(\mathrm{VTC}(\mathbf{z}_t, \mathbf{o}, \mathbf{v}), t)$. Before elaborating on our approach's training and sampling processes, we introduce the proposed VTC module below.

## 3.4 VISION-TEXT CONDITION (VTC) MODULE

To integrate the vision guidance into the text diffusion model, we propose the VTC module to fuse the noisy text features with the visual features of the image and the target object before noise prediction. VTC consists of two modules: the cross-attention module and the token selection module. First, two noisy vision-language features are obtained at the $t$-th step: (i) $\mathbf{z}_t^v = \{z_{i,t}^v\}_{i=1}^{L}$ are computed by cross attention between $\mathbf{z}_t$ and $\mathbf{v}$ to condition on the entire image. (ii) $\mathbf{z}_t^o = \{z_{i,t}^o\}_{i=1}^{L}$ are also computed by cross attention between $\mathbf{z}_t$ and $\mathbf{o}$ to condition on the target object. Next, we introduce a token selection module aimed at enhancing the features associated with the target object within $\mathbf{z}_t^v$. Specifically, we calculate the cosine similarity between the features of $\mathbf{z}_t^v$ and $\mathbf{z}_t^o$, resulting in the following score matrix $S = [s_{ij}] \in \mathbb{R}^{L \times L}$, where the $i$-th row of $S$ denotes the similarities between the $i$-th feature in $\mathbf{z}_t^v$ and all features in $\mathbf{z}_t^o$. We then sum up the scores within each row of $S$ to obtain a set of scores, denoted by $S^r = [s_1^r, ..., s_L^r]^T$. If $s_j^r >= \frac{\sum_{i=1}^{L} s_i^r}{3}$, we consider $z_{j,t}^v$ as highly relevant to the target object and extract it from $\mathbf{z}_t^v$. Suppose there are $k$ noisy vision-language features extracted from $\mathbf{z}_t^v$, and $z_{j,t}^v$ is among these $k$ noisy vision-language features. For $z_{j,t}^v$, we also update the corresponding noisy vision-language feature for the target object as

$$z_{j,t}^s = \sum_{i=1}^{L} \hat{s}_{ji} z_{i,t}^o, \tag{4}$$

where $\hat{s}_{ji} = \frac{\exp(s_{ji})}{\sum_{l=1}^{L} \exp(s_{jl})}$. Finally, the noisy vision-language features $\hat{z}_{j,t}$, resulting from combining both $z_{j,t}^v$ and $\mathbf{z}_t^o$, can be acquired by

$$\hat{z}_{j,t} = \mathbf{W}\left[z_{j,t}^v; z_{j,t}^s\right], \tag{5}$$

where $\mathbf{W}$ is a trainable parameter matrix. Repeat the process in (4) and (5) until all features extracted from $\mathbf{z}_t^v$ are updated. The updated features (*i.e.*, $\hat{z}_{j,t}$) will form $\hat{\mathbf{z}}_t$, which is then fed into the transformer decoder for noise prediction.

## 3.5 TRAINING AND SAMPLING

**Training.** In the training phase, according to the additivity of Gaussian noise, the training process can be simplified as follows. Given the embedding feature $\mathbf{z}_0$ of the expression, we introduce Gaus-

sian noise $\varepsilon \sim \mathcal{N}(0, \mathbf{I})$ to $\mathbf{z}_0$ to acquire the noisy feature $\mathbf{z}_t$ for the $t$-th step in the forward process. The noisy feature $\mathbf{z}_t$, along with the visual features of the image $\mathbf{v}$ and the target object $\mathbf{o}$, are fed into the VTC module to extract the noisy vision-text features. The noisy features are subsequently fed into the transformer decoder for noise prediction. Specifically, the training objective is formulated as

$$
\mathbf{z}_t = \sqrt{\bar{\alpha}_t}\mathbf{z}_0 + \sqrt{1 - \bar{\alpha}_t}\varepsilon,
$$
$$
\mathcal{L}_{REG} = \mathbb{E}_{\mathbf{o},\mathbf{v},\varepsilon \sim \mathcal{N}(0,\mathbf{I}),t}\left[\|\varepsilon - \varepsilon_\theta\left(\text{VTC}\left(\mathbf{z}_t, \mathbf{o}, \mathbf{v}\right), t\right)\|_2^2\right]. \tag{6}
$$

Existing literature Ho & Salimans (2021) shows that classifier-free guidance contributes to improving the balance between diversity and accuracy. This enhancement aids the diffusion model in generating more accurate and diverse outputs. Within the realm of classifier-free guidance, the function $\varepsilon_\theta\left(\text{VTC}\left(\mathbf{z}_t, \mathbf{o}, \mathbf{v}\right), t\right)$ in Eq. 6 can be modified as

$$
\tilde{\varepsilon}_\theta\left(\text{VTC}\left(\mathbf{z}_t, \mathbf{o}, \mathbf{v}\right), t\right) = \varepsilon_\theta\left(\text{VTC}\left(\mathbf{z}_t, \mathbf{o}, \mathbf{v}\right), t\right) + \gamma\left(\varepsilon_\theta\left(\text{VTC}\left(\mathbf{z}_t, \mathbf{o}, \mathbf{v}\right), t\right) - \varepsilon_\theta\left(\mathbf{z}_t, t\right)\right), \tag{7}
$$

where $\gamma$ is the guidance weight. The proportion of samples for unconditional process is set to 0.2. In this case, training objective is formulated as

$$
\tilde{\mathcal{L}}_{REG} = \mathbb{E}_{\mathbf{o},\mathbf{v},\varepsilon \sim \mathcal{N}(0,\mathbf{I}),t}\left[\|\varepsilon - \tilde{\varepsilon}_\theta\left(\text{VTC}\left(\mathbf{z}_t, \mathbf{o}, \mathbf{v}\right), t\right)\|_2^2\right]. \tag{8}
$$

**Sampling.** In the sampling phase, we first randomly sample the initial latent $\mathbf{z}_T \sim \mathcal{N}(0, \mathbf{I})$ and feed $\mathbf{z}_T$ together with the visual features, $\mathbf{v}$ and $\mathbf{o}$, into VTC followed by the transformer decoder for noise prediction to sample $\mathbf{z}_{T-1}$. Then, we repeat the same procedure until reaching $\mathbf{z}_0$ as outlined below:

$$
\mathbf{z}_{t-1} = \frac{1}{\sqrt{\alpha_t}}\left(\mathbf{z}_t - \frac{1 - \alpha_t}{\sqrt{1 - \bar{\alpha}_t}}\varepsilon_\theta\left(\text{VTC}\left(\mathbf{z}_t, \mathbf{o}, \mathbf{v}\right), t\right)\right) + \sigma_t\mathbf{n}, \tag{9}
$$

where $t = T, ..., 1$, $\sigma_t$ represents the standard deviation at the $t$-th denoising step, and $\mathbf{n} \sim \mathcal{N}(0, \mathbf{I})$. In classifier-free guidance, $\mathbf{z}_{t-1}$ is also sampled using Eq. 9, where $\varepsilon_\theta$ is replaced by $\tilde{\varepsilon}_\theta$ in Eq. 7.

Finally, we adopt the rounding operation in Reid et al. (2023) to decode the denoised $\mathbf{z}_0$ into expression. Moreover, we use the Minimum Bayes Risk (MBR) decoding strategy Koehn (2004) to enhance the quality of generation, which generates a set of candidate samples by applying different random seeds to our model. The final output sequence is selected based on the minimum expected risk, as determined by a relevant loss function (*e.g.*, Meteor).

### 3.6 DATASET AUGMENTATION FOR REC

Suppose there are $M$ images in the training set of the REC dataset, and our VIE-DM generates $T$ expressions for each image and target object, with each generated expression assigned a score (e.g., Meteor, CIDEr, or other metric scores) denoting its semantic similarity to the GT expression. Subsequently, the expression with the highest score is selected for each image and its target object, yielding an image-expression pair. The score of the $i$-th image-expression pair is denoted as $g_i$. Next, we sort the $M$ scores $\{g_i\}_{i=1}^M$ and select the top $h$ image-expression pairs to integrate them into the training set, where $h = \lceil 0.3M \rceil$ in this work. Finally, we employ existing transformer-based REC methods (i.e., TransVG, M-DGT, VLTVG, QRNet, MDETR and OFA-base) on both augmented and unaugmented datasets to evaluate the quality and diversity of our generated expressions.

## 4 EXPERIMENTAL RESULTS

In this section, we evaluate the performances of the proposed and existing methods for the REG task. Since the REG task and image captioning share some similarities, we will analyze and present experimental comparisons of these two tasks in the supplementary material. In addition, we integrate our generated expressions into different REC datasets for model training and evaluate the performance gains for the REC task to validate the effectiveness of the proposed approach. The details are described in the following sections.

### 4.1 DATASETS AND EVALUATION METRICS

**Datasets.** We evaluate our method on RefCOCO Kazemzadeh et al. (2014), RefCOCO+ Kazemzadeh et al. (2014), RefCOCOg Mao et al. (2016), Flickr30k enetities Plummer

Table 1: REG performance comparison among VIE-DM and existing REG methods on RefCOCO, RefCOCO+, and RefCOCOg. CFG and MBR represent classifier-free guidance and Minimum Bayes Risk, respectively.

| Model | Features | RefCOCO | | | | RefCOCO+ | | | | RefCOCOg | |
| | | testA | | testB | | testA | | testB | | val-g | |
| | | Meteor | CIDEr | Meteor | CIDEr | Meteor | CIDEr | Meteor | CIDEr | Meteor | CIDEr |
|---|---|---|---|---|---|---|---|---|---|---|---|
| Speaker+MMI (baseline) Mao et al. (2016) | VGG16 | 0.243 | 0.615 | 0.300 | 1.227 | 0.199 | 0.462 | 0.189 | 0.679 | 0.149 | 0.585 |
| Speaker+visdif+MMI Yu et al. (2016) | VGG16 | 0.260 | 0.679 | 0.319 | 1.276 | 0.202 | 0.475 | 0.196 | 0.683 | 0.147 | 0.573 |
| Speaker+listener+reinforcer+MMI Yu et al. (2017) | VGG16 | 0.268 | 0.679 | 0.329 | 1.323 | 0.204 | 0.494 | 0.202 | 0.709 | 0.154 | 0.592 |
| Speaker+attr+MMI Liu et al. (2017) | VGG16 | 0.274 | 0.710 | 0.313 | 1.257 | 0.219 | 0.512 | 0.203 | 0.704 | 0.157 | 0.639 |
| VC W/Gen Niu et al. (2021) | VGG16 | 0.188 | 0.707 | 0.245 | 1.356 | 0.142 | 0.518 | 0.146 | 0.731 | 0.139 | 0.625 |
| Speaker+attr+attn+visdif+MMI Liu et al. (2020) | VGG16 | 0.312 | 0.802 | 0.332 | 1.301 | 0.236 | 0.585 | 0.206 | 0.692 | 0.163 | 0.645 |
| PFOS Sun et al. (2023) | VGG16 | 0.292 | 0.829 | 0.319 | 1.281 | 0.237 | 0.661 | 0.198 | 0.740 | 0.157 | 0.722 |
| **VIE-DM w/o MBR** | VGG16 | 0.358 | 1.069 | 0.383 | 1.762 | 0.366 | 1.147 | 0.393 | 1.276 | 0.492 | 1.127 |
| **VIE-DM w/o CFG** | VGG16 | 0.363 | 1.075 | 0.377 | 1.775 | 0.374 | 1.152 | 0.411 | 1.269 | 0.503 | 1.135 |
| **VIE-DM** | VGG16 | 0.373 | 1.077 | 0.391 | 1.782 | 0.378 | 1.160 | 0.408 | 1.285 | 0.509 | 1.132 |
| Speaker+listener+reinforcer+MMI Yu et al. (2017) | ResNet101 | 0.296 | 0.804 | 0.341 | 1.358 | 0220 | 0.579 | 0.221 | 0.798 | 0.153 | 0.742 |
| Speaker+reinforce+visdif+MMI Tanaka et al. (2019) | ResNet101 | 0.307 | 0.865 | 0.343 | 1.381 | 0.242 | 0.671 | 0.220 | 0.812 | 0.164 | 0.738 |
| Speaker+listener+reinforce+visdif+MMI Tanaka et al. (2019) | ResNet101 | 0.310 | 0.859 | 0.342 | 1.375 | 0.241 | 0.663 | 0.225 | 0.812 | 0.164 | 0.763 |
| PFOS Sun et al. (2023) | ResNet101 | 0.303 | 0.877 | 0.330 | 1.333 | 0.253 | 0.722 | 0.210 | 0.758 | 0.156 | 0.749 |
| MiniGPT-v2 Chen et al. (2023) | ViT | 0.417 | 1.038 | 0.425 | 1.767 | 0.405 | 1.124 | 0.413 | 1.317 | 0.612 | 1.129 |
| CCL Wang et al. (2024) | ViT | 0.348 | 1.042 | 0.379 | 1.566 | - | - | - | - | - | - |
| **VIE-DM w/o MBR** | ResNet101 | 0.432 | 1.187 | 0.448 | 1.966 | 0.439 | 1.516 | 0.474 | 1.652 | 0.694 | 1.535 |
| **VIE-DM w/o CFG** | ResNet101 | 0.439 | 1.193 | 0.463 | **2.085** | 0.446 | 1.538 | 0.485 | **1.675** | 0.712 | 1.546 |
| **VIE-DM** | ResNet101 | **0.445** | **1.207** | **0.472** | 2.014 | **0.453** | **1.543** | **0.491** | 1.663 | **0.735** | **1.558** |

et al. (2015) and Refclef Kazemzadeh et al. (2014) to assess its effectiveness. Due to space limitations, detailed descriptions of these five datasets are provided in the Appendix.

**Evaluation Metrics.** BLEU Papineni et al. (2002), Meteor Lavie & Agarwal (2007), Rouge Lin (2004), and CIDEr Vedantam et al. (2015) are four widely used evaluation metrics for the image captioning and REG. Among them, Meteor and CIDEr take better consideration in terms of synonym matches and sentence structures than BLEU and Rouge. Thus, we employ Meteor and CIDEr to evaluate the generation diversity and quality of the proposed approach. Additionally, we employ metrics Div-1 Shetty et al. (2017), Div-2 Shetty et al. (2017), and mBLEU-4 Shetty et al. (2017) to assess the diversity of our generated expressions. Note that the value range of Meteor, Div-1, Div-2, and mBLEU-4 are between 0 and 1. We follow the standard protocol to evaluate the REC task using the mAP@1 metric. When the Intersection over Union (IoU) between the detected object and the ground truth is greater than 0.5, this prediction is correct.

**Implementation Details.** We employs a VTC module with two independent cross-attention layers followed by a transformer decoder, where the decoder consists of 12 self-attention layers and 12 attention heads for each layer. The established model comprises 1.2 billion parameters. For REG training on the RefCOCO dataset (120,624 image-expression pairs), the network with classifier-free guidance takes approximately 86 hours on 4 NVIDIA V100 GPUs . The inference time using one NVIDIA V100 GPU is 8.74 seconds per image with DDPM and 0.93 seconds per image with DDIM. Like position embedding, time step embedding is incorporated. Furthermore, the maximum sequence length is set to 128, with an embedding dimension $d = 128$, a square-root noise schedule, and 2000 diffusion steps. The batch size and learning rate are set to 8 and 1e-04, respectively. The parameter $\gamma$ is set to 3.0 empirically. The optimizer, vision encoder, and textual encoder employed are AdamW, VIT/B-32, and BERT, respectively. Regarding the accuracy metrics using Minimum Bayes Risk (MBR) Koehn (2004), we utilize a set of 10 candidate samples.

## 4.2 Performance Evaluation Results

We first compare our method with other state-of-the-art REG methods regarding the fidelity of the generated expressions on the RefCOCO, RefCOCO+, and RefCOCOg datasets in Table 1. Note that "visdif" and "reinforcer" are two modules introduced in Yu et al. (2016) and Yu et al. (2017), respectively, while "attr" and "attn" are the other two modules proposed in Liu et al. (2020). With the VGG16 features, all methods are CNN-LSTM-based frameworks except PFOS. We observe a subpar performance in baseline, especially in Meteor, due to the limited model capacity of the CNN-LSTM models. Incorporating the "attr" and "atten" modules better bridges the gap between the visual and textual domains than the vanilla CNN-LSTM models. PFOS is the sole method based on the transformer-LSTM framework. However, its potential for improvement is constrained since it doesn't introduce additional information to bridge the gap between the vision transformer and LSTM. MiniGPT-v2 stands out as an all-in-one approach that delivers impressive performance

Table 2: Experimental results of existing REC methods on the original and augmented RefCOCO, RefCOCO+, RefCOCOg, Flickr30k, and Refclef. Here, CC Sharma et al. (2018), SBU Ordonez et al. (2011), COCO Pont-Tuset & Van Gool (2015), VG Krishna et al. (2017) and Flickr30K Plummer et al. (2015) are the datasets used for pre-training. Better results are marked in bold. CO and F30K represent COCO and Flickr30K, respectively.

| Methods | Pre-train Dataset | RefCOCO | | | RefCOCO+ | | | RefCOCOg | | Flickr30k | Refclef |
|---|---|---|---|---|---|---|---|---|---|---|---|
| | | val | testA | testB | val | testA | testB | val | test | test | test |
| TransVG Deng et al. (2021) | CO | 81.02 | 82.72 | 78.35 | 64.82 | 70.70 | 59.64 | 68.67 | 67.73 | 79.10 | 70.73 |
| **TransVG+VIE-DM** | CO | **83.75** | **84.58** | **81.13** | **67.09** | **72.52** | **62.67** | **70.93** | **69.65** | **81.54** | **72.37** |
| M-DGT Chen & Li (2022) | None | 85.37 | 83.01 | 85.24 | 70.02 | 72.26 | 68.92 | 79.21 | 79.06 | **79.97** | - |
| **M-DGT+VIE-DM** | None | **86.67** | **84.35** | **86.43** | **70.28** | **73.35** | **70.18** | **81.48** | **79.95** | 80.57 | - |
| VLTVG Yang et al. (2022) | CO | 84.77 | 87.24 | 80.49 | 74.19 | 78.93 | 65.17 | 76.04 | 74.18 | 79.84 | 71.98 |
| **VLTVG+VIE-DM** | CO | **86.58** | **88.83** | **82.59** | **76.42** | **80.93** | **67.82** | **78.83** | **76.08** | **81.28** | **73.37** |
| QRNet Ye et al. (2022) | CO | 84.01 | 85.85 | 82.34 | 72.94 | 76.17 | 63.81 | 73.03 | 72.52 | 81.95 | 74.61 |
| **QRNet+VIE-DM** | CO | **86.82** | **88.50** | **84.66** | **75.35** | **77.89** | **66.25** | **74.86** | **74.67** | **83.58** | **76.28** |
| MDETR Kamath et al. (2021) | CO, VG, F30K | 86.57 | 89.58 | 81.41 | 79.52 | 84.09 | 70.62 | 81.64 | 80.89 | 83.80 | - |
| **MDETR+VIE-DM** | CO, VG, F30K | **90.53** | **92.15** | **85.95** | **82.84** | **86.83** | **73.35** | **83.80** | **83.56** | **86.08** | - |
| OFA-base Wang et al. (2022) | VG | 88.48 | 90.67 | 83.30 | 81.39 | 87.15 | 74.29 | 82.29 | 82.31 | - | - |
| **OFA-based+VIE-DM** | VG | **90.53** | **91.75** | **85.63** | **84.08** | **89.65** | **76.20** | **84.27** | **83.98** | - | - |

Table 3: Diversity comparisons among VIE-DM and other REG methods. Larger values mean higher diversity in all metrics except for mBleu-4, where lower values are preferred. SM, SLRVM and SLRM represent Speaker+MMI, Speaker+listener+reinforce+visdif+MMI and Speaker+listener+reinforcer+MMI, respectively.

| Model | RefCOCO | | | | | | RefCOCO+ | | | | | | RefCOCO+ | | |
|---|---|---|---|---|---|---|---|---|---|---|---|---|---|---|---|
| | testA | | | testB | | | testA | | | testB | | | val-g | | |
| | mBLEU-4 | Div1 | Div2 | mBLEU-4 | Div1 | Div2 | mBLEU-4 | Div1 | Div2 | mBLEU-4 | Div1 | Div2 | mBLEU-4 | Div1 | Div2 |
| SM | 0.473 | 0.357 | 0.423 | 0.452 | 0.374 | 0.450 | 0.465 | 0.343 | 0.435 | 0.458 | 0.363 | 0.471 | 0.377 | 0.412 | 0.468 |
| SLRM | 0.576 | 0.338 | 0.393 | 0.554 | 0.353 | 0.407 | 0.533 | 0.298 | 0.355 | 0.529 | 0.330 | 0.408 | 0.485 | 0.363 | 0.392 |
| SLRVM | 0.595 | 0.327 | 0.407 | 0.538 | 0.385 | 0.395 | 0.603 | 0.296 | 0.363 | 0.556 | 0.348 | 0.383 | 0.506 | 0.366 | 0.409 |
| **VIE-DM** | **0.251** | **0.526** | **0.517** | **0.276** | **0.509** | **0.522** | **0.247** | **0.518** | **0.579** | **0.256** | **0.496** | **0.558** | **0.308** | **0.567** | **0.634** |

across various downstream vision-language tasks. As presented in Table 1, our VIE-DM outperforms MiniGPT-v2 in a favorable manner. Moreover, MiniGPT-v2 is based on Llama2-Chat-7B, which includes at least 7B parameters, while VIE-DM only includes about 1.2B parameters. In the section Appendix, we will provide an example that demonstrates a comparison between our method and MiniGPT-v2. In addition, comparing the methods using VGG16 features with those using ResNet101 features, the ResNet101-based approaches yield better performances.

The proposed VIE-DM consistently outperforms the state-of-the-art REG methods utilizing both ResNet101 and VGG16 backbones. Specifically, in testA of RefCOCO, VIE-DM surpasses the Speaker+listener+reinforce+visdif+MMI method with ResNet101 by 0.135 and 0.348 in the Meteor and CIDEr metrics, respectively. When comparing VIE-DM under different conditions, we find that the performances of VIE-DM generally surpass those of VIE-DM without MBR and CFG in most cases. These results demonstrate the high quality of our VIE-DM.

We utilize generated expressions by VIE-DM to augment five REC datasets and apply six transformer-based REC methods to both augmented and unaugmented datasets, aiming to evaluate the diversity and quality of our generated expressions. As reported in Table 2, our method consistently enhances the transformer-based REC methods on these datasets. Notably, even though the method doesn't rely on pre-training (e.g., with M-DGT), the performances across the five augmented datasets remain stable. For the methods pre-trained on additional datasets, there is a significant improvement in performance using the augmented datasets, particularly for MDETR. In most cases, MDETR using the augmented datasets (i.e., MDETR+VIE-DM) outperforms the state-of-the-art transformer-based methods (e.g., OFA-base). These findings further underscore the enhancement in diversity that our generated expressions bring to the original datasets. In addition, they also validate our method's effectiveness and potential to improve other transformer-based REC methods.

We compare our method with other REG methods using metrics Div-1, Div-2, and mBLEU-4 to evaluate the diversity of generated expressions. As given in Table 3, when the "listener" and "reinforcement" modules are added to the baseline Speaker+MMI model, the generated expressions can better match the ground truth but with poor diversity. In contrast, combining these results with those in Table 1, our methods not only produce high-quality expressions but also exhibit strong diversity.

Table 4: VIE-DM for REG without the VTC module or without the token selection strategy.

| Model | RefCOCO | | | | RefCOCO+ | | | | RefCOCOg | |
|---|---|---|---|---|---|---|---|---|---|---|
| | testA | | testB | | testA | | testB | | val-g | |
| | Meteor | CIDEr | Meteor | CIDEr | Meteor | CIDEr | Meteor | CIDEr | Meteor | CIDEr |
| VIE-DM w/o VTC | 0.335 | 0.956 | 0.379 | 1.512 | 0.383 | 1.308 | 0.376 | 1.318 | 0.417 | 1.382 |
| VIE-DM w/o token selection strategy | 0.426 | 1.163 | 0.450 | 1.907 | 0.426 | 1.486 | **0.503** | 1.592 | 0.702 | 1.502 |
| **VIE-DM** | **0.445** | **1.207** | **0.472** | **2.014** | **0.453** | **1.543** | 0.491 | **1.663** | **0.735** | **1.558** |

Table 5: Experimental results of QRNet on the augmented datasets by including 10%, 30%, 50%, and 70% of the generated image-expression pairs into the training sets.

| Amount of augmented pairs | | | | RefCOCO | | RefCOCO+ | | RefCOCOg |
|---|---|---|---|---|---|---|---|---|
| 10% | 30% | 50% | 70% | testA | testB | testA | testB | test |
| ✓ | | | | 86.75 | 83.83 | 77.58 | 65.03 | 73.35 |
| | ✓ | | | **88.50** | 84.66 | **77.89** | 66.25 | **74.67** |
| | | ✓ | | 85.53 | **85.82** | 75.60 | 63.78 | 70.25 |
| | | | ✓ | 84.98 | 85.23 | 77.85 | 62.93 | 70.65 |

Table 6: The results of QRNet are shown for two scenarios: randomly selecting 30% of image-expression pairs (RS), and sorting and selecting the top 30% of image-expression pairs (SS)

| case | | RefCOCO | | RefCOCO+ | | RefCOCOg |
|---|---|---|---|---|---|---|
| RS | SS | testA | testB | testA | testB | test |
| ✓ | | 87.14 | 83.88 | **78.03** | 65.12 | 73.54 |
| | ✓ | **88.50** | **84.66** | 77.89 | **66.25** | **74.67** |

## 4.3 ABLATION STUDIES

In the following, we use RefCOCO, RefCOCO+, and RefCOCOg to conduct the ablation studies. We first perform unconditional generation without employing the vision-text condition module by directly feeding the noisy features into the transformer decode for noise prediction and sampling. As shown in the first row of Table 4, the performances of VIE-DM without the VTC module are noticeably lower than those with the VTC module. These results indicate that, with the guidance of images, VIE-DM can generate more accurate and diverse expressions for the target objects.

Then, we validate the effectiveness of the token selection strategy exploited by the VTC module. As shown in the second row of Table 4, when the proposed VIE-DM takes all tokens from the image into consideration without performing this strategy, it results in inferior performances to those using this strategy in most cases. This outcome demonstrates that our proposed method can effectively select the tokens from the image containing more relevant information about the target objects.

To evaluate the efficacy of the augmented dataset by varying the number of REG-augmented samples, we randomly sample 10%, 30%, 50%, and 70% of image-expression pairs generated by VIE-DM and incorporate them into the training sets of the original REC dataset. We also employ QRNet as the baseline and evaluate its performance on the four augmented datasets. The results in Table 5 indicate a progressive improvement in the performances of QRNet when 10% and 30% of the augmented image-expression pairs are incorporated. The performance hits the plateau when more than 30% of the generated image-expression pairs are included. We attribute this to two factors: first, the diversity of expressions already reaches a sufficient level, and second, higher augmentation percentages introduce potentially inaccurate pairs, which negatively impact the method.

We also evaluate the efficacy of the augmented dataset generated by VIE-DM in two scenarios. First, we randomly select 30% of image-expression pairs from VIE-DM and integrate them into the training sets of the original REC dataset. Second, we sort and select the top 30% of image-expression pairs with higher Meteor scores and integrate them into the original REC dataset. As depicted in Table 6, we observe that the performances of QRNet on the augmented dataset are superior in the second case compared to the first one. These findings not only demonstrate the rich diversity of expressions generated by VIE-DM but also indicate that the expressions it generates, especially those with higher Meteor scores, provide more accurate descriptions of the target objects.

## 4.4 VISUALIZATION

In addition to presenting quantitative results, we also provide visual illustrations of our method in Figure 3. The first and second rows exhibit the accurate outcomes yielded by our method, while the third row illustrates instances with lower Meteor scores. Spcifically, the first two rows illustrate the effectiveness of our method in generating accurate and diverse expressions, where the noun phrases in the original expressions are usually replaced with synonyms while the sentences are sometimes restructured. Among the observed patterns, the faithful image-text relationships can be attributed to the proposed VTC (vision-text condition) module, which ensures precise alignment between visual and textual tokens. Meanwhile, the transformer decoder and diffusion model contribute to high

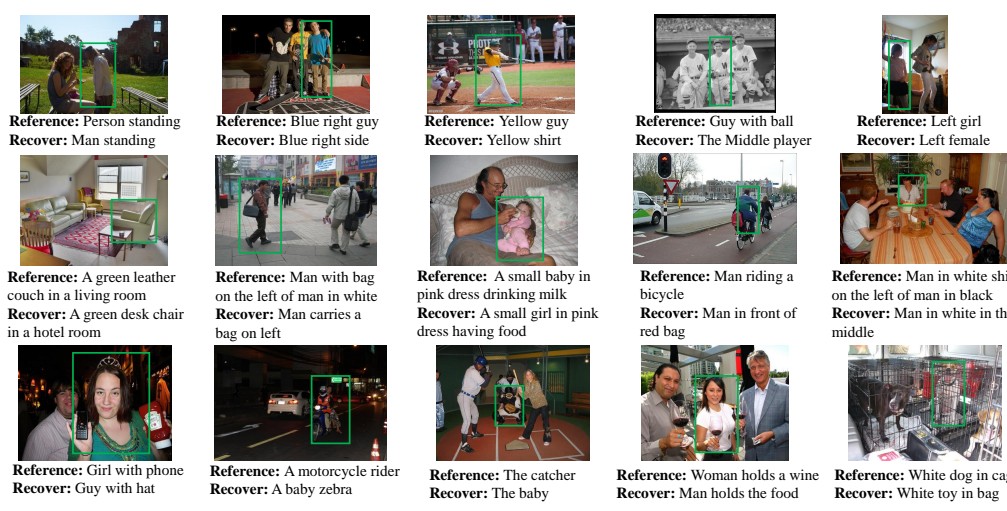

Figure 3: Visualization of VIE-DM. The first and second rows show the accurate results of VIE-DM in short and long descriptions, while the third row shows the inaccurate results.

diversity through synonym replacement and sentence restructuring. Regarding the first sample of the third row, although our VIE-DM results in an expression ("Girl with hat.") different from the reference ("Girl with phone."), it does capture the phenomenon that the girl wears something. For the second sample, since the stripe pattern on the shirt worn by the rider shares some resemblance with the zebra, VIE-DM makes an inaccurate expression generation. Lastly, in the third sample, due to the small patch size providing limited information, the proposed approach results in an erroneous generation. While our method may generate some inaccurate expressions, we have implemented a mechanism to effectively eliminate these inaccuracies by utilizing metrics such as METEOR during the process of augmenting the REC dataset.

## 4.5 DISCUSSION

In dataset augmentation, some studies such as Yuan et al. (2024); Zhang et al. (2023) have successfully employed diffusion models. Nevertheless, using diffusion models in vision-guided expression generation for the REG and REC tasks remains unexplored and non-trivial. Specifically, Yuan et al. (2024); Zhang et al. (2023) and other diffusion-based data augmentation methods do not explicitly handle the misalignment issues between the vision and language modalities for the REG task. However, our proposed vision-text condition (VTC) with the token selection strategy effectively addresses the critical issue of the REG task. It helps the model generate expressions that faithfully capture the attributes of the target object in the image. As shown in Table 4 of the paper, neglecting the proposed VTC or the token selection strategy in the diffusion model suffers from substantial performance drops, which can be observed consistently on all five datasets in both Meteor and CIDEr.

## 5 CONCLUSION

In this paper, to the best of our knowledge, we make a pioneering contribution by introducing the diffusion models for referring expression generation. Our proposed **VI**sion-guided **E**xpression **D**iffusion **M**odel (VIE-DM) demonstrates its effectiveness in generating accurate and diverse expressions for target objects within a given image. In VIE-DM, by incorporating a vision-text condition module, we can efficiently extract visual tokens from the entire image that are relevant to the target object and fuse them with the noisy text features. This integration aids in generating more accurate expressions for the target object in the image. Additionally, we illustrate that expressions generated by our method significantly enhance the REC dataset. This augmented REC dataset, when utilized by state-of-the-art transformer-based REC methods, leads to improved performance. With these distinct advantages, our method emerges as a powerful tool for the REG and REC tasks.

## 6 ACKNOWLEDGEMENT

This work was supported in part by the National Science and Technology Council (NSTC) under grants 112-2221-E-A49-090-MY3, 112-2222-E-001-001-MY2, 112-2221-E-007-077-MY3, and 113-2634-F-002-003. This work was funded in part by Academia Sinica under the grant number of AS-CDA-110-M09 and by MediaTek. The authors also thank National Center for High-performance Computing (NCHC) of National Applied Research Laboratories (NARLabs) in Taiwan for providing computational and storage resources.

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

## A  APPENDIX

### A.1  DATASETS

In the following, we provide a concise overview of the datasets utilized in our studies for both training and evaluation purposes.

**RefCOCO**, **RefCOCO+**, and **RefCOCOg:** The RefCOCO and RefCOCO+ datasets comprise 142,210 and 141,564 expressions about 50,000 and 49,856 objects, across 19,994 and 19,992 images, respectively. In both datasets, expressions are gathered through an interactive game interface. Specifically, as described in Kazemzadeh et al. (2014), RefCOCO is divided into four sets: training, validation, testA, and testB. They contain 120,624, 10,834, 5,657, and 5,095 samples, respectively. The testA dataset primarily focuses on images with multiple people, while testB encompasses images with various objects. Similarly, RefCOCO+ follows the same four-split configuration, with 120,191, 10,758, 5,726, and 4,889 samples for the training, validation, testA, and testB, respectively. In contrast to RefCOCO, the expressions in RefCOCO+ do not include descriptions of absolute location. For RefCOCOg, it is collected in a non-interactive setting, featuring 95,010 lengthy expressions corresponding to 49,822 objects within 25,799 images. This dataset includes 80,512, 4,896, and 9,602 image-expression pairs for the training, validation, and test sets, respectively. Notably, the RefCOCO, RefCOCO+, and RefCOCOg datasets are derived from the MSCOCO dataset Pont-Tuset & Van Gool (2015), which encompasses 80 object categories.

Table 7: Quantitative comparison of REG performances of VIE-DM with DDIM and DDPM, respectively.

| Model | RefCOCO | | | | RefCOCO+ | | | | RefCOCOg | |
|---|---|---|---|---|---|---|---|---|---|---|
| | testA | | testB | | testA | | testB | | val-g | |
| | Meteor | CIDEr | Meteor | CIDEr | Meteor | CIDEr | Meteor | CIDEr | Meteor | CIDEr |
| VIE-DM (w/ DDIM) | 0.435 | **1.213** | 0.467 | 1.984 | 0.457 | 1.502 | 0.485 | 1.642 | 0.707 | 1.533 |
| **VIE-DM (w/ DDPM)** | **0.445** | 1.207 | **0.472** | **2.014** | **0.453** | **1.543** | **0.491** | **1.663** | **0.735** | **1.558** |

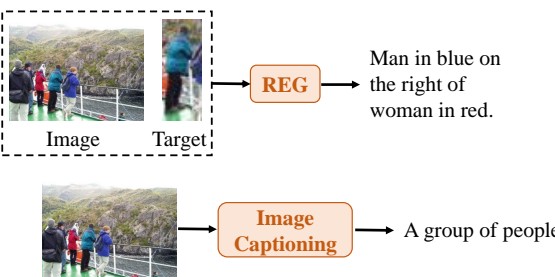

Figure 4: Visualization of REG and image captioning.

**Flickr30K:** The dataset comprises 31,783 images, each accompanied by five descriptive sentences. Each sentence refers to objects belonging to one of seven common object categories and an "other" category for objects not falling under these seven categories. The dataset is partitioned into 425,831 training samples, 14,433 validation samples, and 14,481 test samples.

**RefClef:** The dataset is gathered using ReferItGame, featuring 19,997 images sourced from the SAIAPR-12 dataset. It encompasses 54,127 training, 5,842 validation, and 60,103 test referring expressions.

## A.2 ADDITIONAL ABLATION STUDIES OF OUR METHOD

As mentioned in Song et al. (2021), diffusion models working with DDIM typically achieve competitive performance while exhibiting higher efficiency with significantly reduced diffusion steps. To evaluate the performances of our diffusion model with DDIM, during the sampling phase, we substitute DDIM for DDPM for the denoising process, configuring the number of diffusion steps in DDIM to 500. The findings presented in Table 7 demonstrate that the performance of VIE-DM with DDIM is comparable to that with DDPM. Considering the number of diffusion steps, it is evident that VIE-DM with DDIM exhibits higher efficiency.

## A.3 COMPARISON BETWEEN REG AND IMAGE CAPTIONING

In REG, the input includes an image and a specific region of interest, while in image captioning, only the image is provided as input. REG generates expressions describing the highlighted region, whereas image captioning generates expressions describing the entire image. The visualization of the difference between the two tasks is shown in Figure 4.

## A.4 DIVERSITY COMPARISON BETWEEN OUR METHOD AND EXISTING REG METHODS

To further gauge the data diversity introduced by different REG methods, we compare the REC performances of the proposed method with other REG methods where we employ QRNet as the baseline and evaluate its performance on the four augmented datasets. These four augmented datasets are derived from Speaker+listener+reinforce+visdif+MMI Tanaka et al. (2019), Speaker+listener+reinforcer+MMI Yu et al. (2017), and our VIE-DM, respectively. Specifically, we collect 30% of samples from the image-expression pairs of these four methods and integrate them into the original datasets for augmentation. The results are reported in Table 8. Combined with the findings from Table 1 and Table 3 in the paper, we observe that the performance of QRNet on the augmented datasets of Speaker+MMI shows a slight decrease, while the performances re-

Table 8: REC performance comparison of QRNet on four kinds of augmented datasets. The best results are marked in bold. SM, SLRVM and SLRM represent Speaker+MMI, Speaker+listener+reinforce+visdif+MMI and Speaker+listener+reinforcer+MMI, respectively.

| Methods | Backbone | RefCOCO | | | RefCOCO+ | | | RefCOCOg | |
|---|---|---|---|---|---|---|---|---|---|
| | | val | testA | testB | val | testA | testB | val | test |
| QRNet Ye et al. (2022) | ResNet101 | 84.01 | 85.85 | 82.34 | 72.94 | 76.17 | 63.81 | 73.03 | 72.52 |
| QRNet+SM Mao et al. (2016) | ResNet101 | 83.35 | 83.87 | 81.56 | 72.28 | 76.83 | 63.53 | 72.76 | 71.28 |
| QRNet+SLRVM Tanaka et al. (2019) | ResNet101 | 84.46 | 86.08 | 82.57 | 73.16 | 76.84 | 63.25 | 72.85 | 72.84 |
| QRNet+SLRM Yu et al. (2017) | ResNet101 | 83.85 | 86.04 | 82.75 | 72.85 | 75.82 | 62.83 | 73.53 | 72.08 |
| **QRNet+VIE-DM** | ResNet101 | **86.82** | **88.50** | **84.66** | **75.35** | **77.89** | **66.25** | **74.86** | **74.67** |

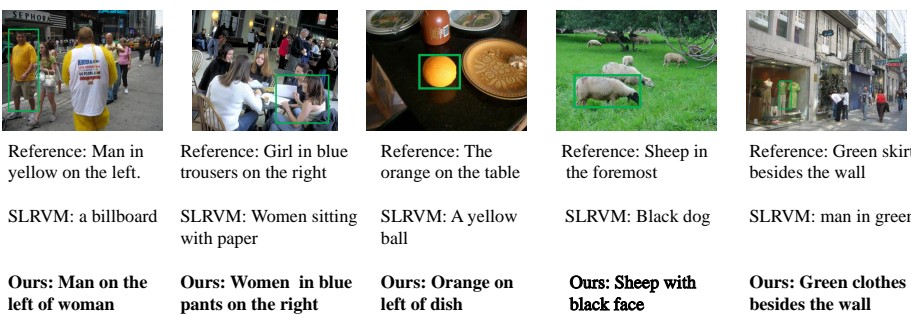

Figure 5: Qualitative comparisons between the proposed VIE-DM (Ours) and SLRVM.

main stable on the other two augmented datasets based on Speaker+listener+reinforce+visdif+MMI and Speaker+listener+reinforcer+MMI. However, on average, the performances of QRNet using the augmented datasets by our VIE-DM exhibit an improvement of approximately 1.5%. These results demonstrate that our VIE-DM significantly enhances the data diversity while keeping the fidelity of the original dataset.

## A.5 COMPARISON BETWEEN OUR REG METHOD AND SLRVM FOR EXPRESSION GENERATION

In the following, we present visual comparisons between our method and existing REG methods. We choose to benchmark our approach against Speaker+Listener+Reinforce+Visdf+MMI (SLRVM) Tanaka et al. (2019), given its superior performance among existing REG methods with the provided source codes. The results are shown in Figure 5. In the first column, our method consistently generates more accurate and diverse descriptions for the target objects, compared to SLRVM, which focuses on the man's back and ultimately results in an inaccurate description. Similarly, in the fourth column, our approach excels in providing more precise and diverse descriptions for the target objects, while SLRVM erroneously identifies the sheep in front as a dog. Furthermore, in the fifth column, SLRVM incorrectly interprets the prop with a green skirt as the man in green.

## A.6 EFFECTIVENESS OF THE DIFFUSION MODEL OF OUR METHOD

To evaluate the importance of the diffusion model in this work, we develop a variant of our method. In this simplified variant, the embedded image and target object are directly fed into the token selection strategy, and the outputs are passed to the transformer decoder. As shown in Table 9, performance decreases significantly without the diffusion model. This underscores the importance of our proposed vision-guided diffusion model for generating high-quality expressions suitable for REC dataset augmentation.

## A.7 VISUALIZATION BETWEEN OUR REG METHOD AND MINIGPT-V2 ON EXPRESSION GENERATION

In this subsection, we present a quality comparison between MiniGPT-v2 and our method for expression generation. In Figure 6, we observe notable differences in the generated sentences. On the left side, MiniGPT-v2 produces an ambiguous sentence as it solely focuses on extracting the

Table 9: Comparison between our method with and without diffusion model on the RefCOCO dataset.

| Model | RefCOCO | | | |
|---|---|---|---|---|
| | testA | | testB | |
| | Meteor | CIDEr | Meteor | CIDEr |
| our method w/o Diffusion | 0.335 | 0.901 | 0.347 | 1.451 |
| **VIE-DM** | **0.445** | **1.207** | **0.472** | **2.014** |

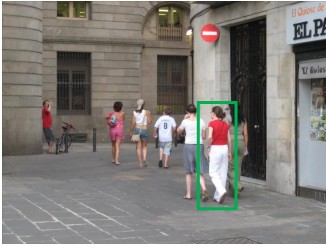 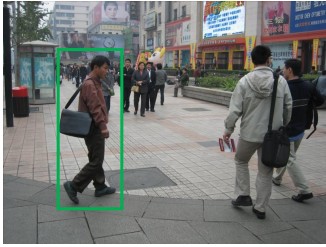

MiniGPT-v2 : A person in red clothes.
VIE-DM: Woman in red between two women.

MiniGPT-v2 : Man with a bag.
VIE-DM: Man carries a bag on left.

Figure 6: Results between MiniGPT-v2 and our VIE-DM.

target region for expression, disregarding the relationship between the target object and other objects in the image. Conversely, our method takes into account both the target object and the entire image simultaneously. This approach enables our method to generate more precise expressions for target object identification. Similarly, on the right side of Figure 6, we observe that our method outperforms MiniGPT-v2 by generating more accurate expressions for target object identification. The expression generated by MiniGPT-v2 appears to be ambiguous, as it fails to clearly indicate which object it is referring to. In contrast, our method produces precise descriptions that specifically identify the target object with accuracy.

