# OpenReview forum: "Generation and Comprehension Hand-in-Hand: Vision-guided Expression Diffusion for Boosting Referring Expression Generation and Comprehension"
_ICLR.cc/2025/Conference — ICLR 2025 Poster_

### Official Review · Reviewer_XHYa · 2024-10-31

**Soundness:** 2
**Presentation:** 3
**Contribution:** 2
**Rating:** 6
**Confidence:** 4

**Summary:**

Existing REC datasets often contain insufficient semantic pairs for training, hindering the REC model's generalization to unseen referring expressions. Additionally, REG methods, due to limited capacity, frequently struggle to bridge the visual and textual domains, resulting in low quality and diversity of generated expressions. In this work, the authors introduce diffusion models into the referring expression generation task, aligning visual features of varying granularity with noisy text. The experiemts are conduted on benchmark datasets.

**Strengths:**

The method is described in detail, and the motivations are fairly well-founded.

**Weaknesses:**

1. Some recent representative works, such as CCL[1], are not compared, even though these works use similar ideas to enhance REC performance through REG.
[1] Cycle-Consistency Learning for Captioning and Grounding. AAAI 2024.
2. Failure cases are lacking; diffusion data generation is usually unstable, and the authors need to analyze this point.
3. Statistics on model parameters, training time, and inference time are required.

**Questions:**

1. Why does performance on certain metrics improve after losing CFG in Table 1?
2. What is the number of samples in each augmented dataset? This needs to be reported.

---

> ### Author Response · Authors · 2024-11-24
>
> We appreciate your insightful comments and have addressed them as follows.
>
> **[Q1] Some recent representative works, such as CCL [Wang et al. AAAI'24], are not compared.**
> A comparison between our method and CCL, presented in the table below, demonstrates the superiority of our method. While CCL generates expressions based solely on predicted objects, our method incorporates a holistic perspective via the proposed VTC (vision-text condition) module, considering both the target object and the surrounding image context. This enables our method to generate more accurate and unambiguous expressions, as illustrated by the example in Figure 6 of the supplemental materials. The comparison with CCL has been included in Table 1 of the revised paper.
>
> |Method|RefCOCO|RefCOCO|
> |-|-|-|
> ||TestA|TestB|val|
> ||Meteor, CIDEr|Meteor, CIDEr|
> |CCL|$0.348$, $1.042$|$0.379$, $1.566$|
> |VIE-DM|$0.445$, $1.207$|$0.472$, $2.014$|
>
> **[Q2] Failure cases are lacking; diffusion data generation is usually unstable, and the authors need to analyze this point.**
> We would like to clarify that failure cases are provided in the original paper. Some inaccurate expressions generated by our method are shown in the third row of Figure 3. An analysis of these failures and a feasible solution for augmentation are given in Lines 508-515.
>
> We acknowledge the potential instability of diffusion-based generation in the original paper. To address this, we propose a strategy for selecting more stable expressions for augmentation, in Lines 469-476 of the original paper, and evaluate it in Table 6. By using this strategy, the selected augmented data consistently and significantly improve six state-of-the-art REC methods across multiple datasets, as shown in Table 2.
>
> **[Q3] Statistics on model parameters, training time, and inference time are required.**
> The established model comprises 1.2 billion parameters. Training on the RefCOCO dataset (120,624 image-expression pairs) with classifier-free guidance takes approximately 86 hours on 4 NVIDIA V100 GPUs. The inference time on a single NVIDIA V100 GPU is 8.74 seconds per image with DDPM and 0.93 seconds per image with DDIM. These details have been included in Lines 357-360 of the revised paper.
>
> **[Q4] Why does performance on certain metrics improve after losing CFG in Table 1?.**
> In CFG, the guidance weight significantly influences the diversity of expressions generated by our method. A higher guidance scale enhances the accuracy of the generated expressions but reduces their diversity, vice versa. In this paper, we set the guidance weight to 0.2 to enhance the diversity of expressions generated by our method, accepting a slight decrease in accuracy as a trade-off.
>
> **[Q5] What is the number of samples in each augmented dataset? This needs to be reported.**
> Thank you for the comment. We have described how to select augmented data and how to determine the number of augmented data in Section 3.6 of the original paper. During augmentation, all training samples from the REC dataset were input into our method to generate expressions. For example, the RefCOCO training set contains 120,624 samples, resulting in the generation of an equal number of image-expression pairs. Among these, 30% of the pairs with the highest scores (i.e., 36,187 pairs) were selected for augmentation. Since the size of the training sets varies across REC datasets, the number of generated samples also varies accordingly.

---

> > ### Author Response · Authors · 2024-12-03
> >
> > Thank you very much for raising our rating!

---

> ### Author Response · Authors · 2024-11-25
> **We look forward to receiving your further feedback on our responses.**
>
> We have made every effort to address your concerns and sincerely hope our responses meet your expectations. We would greatly appreciate your feedback on our replies to help us further enhance the manuscript.

---

> > ### Author Response · Authors · 2024-11-29
> >
> > Dear Reviewer,
> >
> > We have provided detailed responses to your comments and hope they address your concerns. We look forward to receiving your further feedback on our replies.

---

### Official Review · Reviewer_yjk1 · 2024-11-02

**Soundness:** 3
**Presentation:** 4
**Contribution:** 3
**Rating:** 8
**Confidence:** 3

**Summary:**

This paper introduces the Vision-guided Expression Diffusion Model (VIE-DM) to address limitations in referring expression generation (REG) and comprehension (REC) tasks, particularly the scarcity and low diversity of image-expression pairs in existing datasets. The model includes a vision-text condition (VTC) module and a token selection mechanism to mitigate feature discrepancies between the visual and textual domains.

**Strengths:**

1. Introducing a diffusion model to REG is innovative. VIE-DM generates diverse, high-quality synonymous expressions that align with both the visual and textual context of target objects, enriching REC datasets.
2. The experimental design is well-structured, including ablation studies. Extensive experiments on five datasets demonstrate significant improvements in REC and REG model performance, achieving state-of-the-art results.
3. The paper is clearly written and easy to follow.

**Weaknesses:**

No obvious disadvantages were seen.
Like any research work, this paper likely has its own limitations, though they are not explicitly discussed. Including a section on potential limitations would provide a more balanced perspective.

**Questions:**

See weakness.

---

> ### Author Response · Authors · 2024-11-24
>
> We appreciate your insightful comments and have addressed them as follows.
>
> **[Q1] No obvious disadvantages were seen. Including a section on potential limitations would provide a more balanced perspective.**
> Thank you for appreciating this paper. Two limitations of our method are pointed out in the paper. The first limitation is that our method occasionally generates inaccurate expressions. Some inaccurate examples are shown in the third row of Figure 3 of the paper. An analysis of these failures and a feasible solution for augmentation are given in Lines 508-515. Another limitation, shown in Table 5, is that augmenting too many expressions generated by our method results in performance drops. The corresponding discussion is given in Lines 461-468.

---

> > ### Comment · Reviewer_yjk1 · 2024-11-25
> >
> > Thanks for your response. I keep my postive rating.

---

> > > ### Author Response · Authors · 2024-11-25
> > >
> > > Thank you very much for your support!

---

### Official Review · Reviewer_5XtB · 2024-11-03

**Soundness:** 3
**Presentation:** 2
**Contribution:** 3
**Rating:** 6
**Confidence:** 4

**Summary:**

This paper addresses referring expression generation (REG) and referring expression comprehension (REC). In particular, the paper proposes a method for REG that utilizes a language model with a diffusion model and experimental results for REC that augment the dataset with the REG method. The experiments are performed on five representative datasets, three RefCOCOs, Flickr30k, and Refclef, and show that the accuracy of the proposed method for REG is better than the existing methods and that the augmentation of the dataset by the proposed method contributes to multiple REC methods.

**Strengths:**

- To the best of the reviewer's knowledge, this is the first study to introduce language models using diffusion models into REG and REC.
- The proposed method has been evaluated using multiple datasets and multiple ablation studies, and has shown a certain degree of effectiveness.

**Weaknesses:**

- The proposed method is composed of a straightforward combination of existing methods. The Cross-Attention and Token Selection Strategy that make up the proposed method, Vision-Text Condtion, are known to the community, and the Minimum Bayes Risk (MBR) and classifier-free guidance (CFG) that are ablated in Table 1 are not newly proposed in this paper. (In addition, the REG performance of VIE-DM w/o CFG is reported as an ablation study, but the author forgot to explain what CFG stands for, so it is only the reviewer's guess that CFG means classifier-free guidance.)
- As mentioned in the Introduction, the existing methods compared in this paper adopt the transformer-LSTM or CNN-LSTM framework. In other words, the proposed method differs from other methods not only in that it formulates a language model using a diffusion model, but also in that it uses a transformer-based decoder. It is not clear how the combination of a transformer-based decoder and diffusion model outperforms only a transformer-based decoder. Without this comparison, it is not possible to show the effect of introducing a diffusion model into REG.
- In line 416, it is claimed that “These results demonstrate the robust data diversity and quality of our VIE-DM.” However, it is misleading to make such a claim of diversity based on Table 1, which discusses the similarity to the ground truth using Meteor and CIDEr. The argument regarding Table 3 is more convincing, so this claim should have been made elsewhere.
- As the authors acknowledge, Table 5 shows that augmenting the REC data using VIE-DM only leads to a limited improvement due to the accuracy of the synthesized expressions. Therefore, it is essential to show whether VIE-DM is superior to existing approaches. On the other hand, it is not clear how much accuracy is improved when the data set is augmented using methods other than VIE-DM. The idea of amplifying REC data using REG methods has already existed since [Mao+, CVPR 2016]. If it is not possible to show how much REC accuracy is improved when expressions are augmented using methods other than VIE-DM, it will not be possible to understand the extent to which this paper contributes to REC.

**Questions:**

The reviewer would like to receive responses from the authors about the weaknesses.

---

> ### Author Response · Authors · 2024-11-24
>
> We appreciate your insightful comments and have addressed them as follows.
>
> **[Q1] The proposed method is composed of a straightforward combination of existing methods.**
> We acknowledge that some existing techniques have been adapted and used in our proposed method. However, to the best of our knowledge, this work is the first to apply conditional text diffusion models to the REG task. Moreover, it seamlessly integrates the complementary tasks of REG and REC. Additionally, we have developed effective components specifically for REG. For example, the proposed Vision-Text Condition (VTC) module aligns our diffusion-based approach with the REG task, significantly improving its performance. The token selection strategy within the VTC module is designed to mitigate the negative impact of abundant and irrelevant image tokens, a common challenge in REG.
>
> **[Q2] The author forgot to explain what CFG stands for.**
> We appreciate you pointing out this oversight. CFG is an acronym for classifier-free guidance. We have included the definitation in Lines 325-326 of the revised paper.
>
> **[Q3] It is not clear how the combination of a transformer-based decoder and diffusion model outperforms only a transformer-based decoder.**
> We appreciate this comment and conducted an experiment, where the diffusion model is removed from our method VIE-DM. In this simplified variant, the embedded image and target object are directly fed into the token selection strategy, and the outputs are passed to the transformer decoder. As shown in the table below, performance decreases significantly without the diffusion model. This underscores the importance of our proposed vision-guided diffusion model for generating high-quality expressions suitable for REC dataset augmentation. In the revised paper, this experiment has been included in Table 9 and Lines 801-809 of supplementary materials.
>
>
> |Method|RefCOCO|RefCOCO|
> |-|-|-|
> ||TestA|TestB
> ||Meteor,CIDEr|Meteor,CIDEr|
> |our method w/o Diffusion|$0.335, 0.901$|$0.347, 1.451$|
> |VIE-DM|$0.445, 1.207$|$0.472$, $2.014$|
>
> **[Q4] It is misleading to make such a claim of diversity based on Table 1. ... The argument regarding Table 3 is more convincing.**
> Good catch. In the revised paper, we have removed the claim of diversity based on Table 1 and now claim the advantages of high diversity based on the results reported in Table 3.
>
> **[Q5] Table 5 shows that augmenting the REC data using VIE-DM only leads to a limited improvement.**
> We would like to clarify that what Table 5 presents is the performance of an existing REC method QRNet working with different amounts of augmented data generated by VIE-DM, instead of with or without augmented data. The effect of augmenting the REC data using VIE-DM is reported in Table 2, where our method VIE-DM consistently and substantially improves six powerful REC methods across five benchmark datasets.
>
> **[Q6] Meanwhile, it is not clear how much accuracy is improved when the data set is augmented using methods other than VIE-DM.**
> Table 8 in the appendix of the original paper presents a comparison of our VIE-DM method with four existing REG methods in terms of their ability to augment REC datasets. The results demonstrate that VIE-DM more effectively enhances the REC datasets, enabling existing REC methods to achieve better performance.

---

> > ### Comment · Reviewer_5XtB · 2024-11-24
> >
> > - Regarding the response to Q1, there were no changes from the initial review comments.
> > - Q2, Q3, and Q4 have been resolved.
> > - Regarding the comment identified as Q5, the reviewer already knew that Table 5 was a comparison of augmented datasets of different sizes. What the reviewer meant was that the accuracy improvement plateaus at 30%. The comment, “limited improvement,” refers to this point. On the other hand, the comparison with other augmentation methods, which was the main part of this comment, has been resolved in the response to the comment identified as Q6.
> >
> > Overall, the reviewer judged that the effectiveness of the proposed method has been clearly demonstrated in the responses from the authors, although there are still no major changes to the points raised in Q1. The resulting improvement from an initial score of 5 to 6 weakly supports the acceptance of this paper.

---

> ### Author Response · Authors · 2024-11-25
>
> Thank you very much for raising the rating of our paper!
>
> We agree that while REG-based expression augmentation can enhance REC performance, the improvement tends to plateau once the diversity of the augmented image-expression pairs reaches a sufficient level (e.g., an increase of 30% as shown in Table 5). We have clarified this point further in the revised manuscript.

---

### Official Review · Reviewer_jswi · 2024-11-04

**Soundness:** 3
**Presentation:** 2
**Contribution:** 3
**Rating:** 6
**Confidence:** 3

**Summary:**

The paper explores the integration of referring expression generation (REG) and comprehension tasks. To address challenges such as the scarcity of image-expression pairs in training data for REC and the limitations of the REG methods in bridging visual and textual domains, the authors propose a novel vision-guided expression diffusion model for REG. Extensive experiments demonstrate that the proposed method produces high-quality and diverse generated data.

**Strengths:**

1. The paper explores the potential of applying diffusion models to the REG task, an area that has been largely underexplored.
2. The authors introduce a vision-text conditioning module and a token selection strategy, which significantly enhance the alignment between visual and textual information.
3. Extensive experiments and ablation studies validate the generalization capability and effectiveness of the proposed method’s design choices.

**Weaknesses:**

1. In the visualization results shown in Figure 3, the response labeled as “recover” in the first sample of the third row appears to be an error, as does the response in the last sample of the same row. These results indicate that while the current method enhances diversity, it still includes some erroneous responses. How do you ensure the quality of the generated responses?
2. It is intriguing that the ViT backbone of CLIP is considered as a unified vision encoder in MLLM. Could this architecture produce different patterns and further improve performance?
3. The definition of CFG is missing in Table 1.
4. While the paper provides extensive interpretation of the experimental results, it lacks an in-depth analysis of the reasons behind the observed patterns in the results.
5. The writing is somewhat verbose. For instance, in Subsection 3.6, the second sentence is redundant as it repeats the information in the first sentence.
6. Some equations could be improved; for example, Equations 9 and 10 differ by only one symbol.
7. There are a few typos, such as a missing period on line 82 and an incorrect number on line 360.

**Questions:**

See weakness.

---

> ### Author Response · Authors · 2024-11-24
>
> We appreciate your insightful comments and have addressed them as follows.
>
> **[Q1] The generated expressions in the third row of Figure 3 are inaccurate. While the current method enhances diversity, it still includes some erroneous responses. How do you ensure the quality of the generated responses?**
> The third row of Figure 3 does indeed display some inaccurate expressions generated by our method. We acknowledge that our method may occasionally produce incorrect responses, as illustrated in the third row of Figure 3. However, in most cases, it successfully generates diverse and accurate expressions, as demonstrated in the top two rows of the same figure.
>
> As detailed in Lines 303-315 of the main paper, we present a strategy to estimate the quality of generated expressions. Specifically, we leverage Meteor scores to assess images and their corresponding generated expressions, selecting those with high scores as augmented data. As shown in Table 2, the selected augmented data by our method consistently and significantly improves six state-of-the-art REC methods across five datasets.
>
> **[Q2] The ViT backbone of CLIP is considered as a unified vision encoder in MLLM. Could this architecture produce different patterns and further improve performance?**
> We recognize that the ViT backbone of CLIP is a powerful vision encoder for multimodal LLMs due to its ability to effectively align visual features with text features. As reported in Table 1 of the main paper, the competing method MiniGPT-v2, which utilizes ViT, achieves promising results. Thus, we believe that incorporating ViT or other CLIP vision encoders can be beneficial, and we will evaluate our method with different CLIP vision encoders in the paper.
>
> **[Q3] The definition of CFG is missing in Table 1.**
> We appreciate you pointing out this oversight. CFG is an acronym for classifier-free guidance. We have included the definitation in Lines 325-326 of the revised paper.
>
> **[Q4] This paper lacks an in-depth analysis of the reasons behind the observed patterns in the results.**
> The first two rows of Figure 3 illustrate the effectiveness of our method in generating accurate and diverse expressions, where the noun phrases in the original expressions are usually replaced with synonyms while the sentences are sometimes restructured. Among the observed patterns, the faithful image-text relationships can be attributed to the proposed VTC (vision-text condition) module, which ensures precise alignment between visual and textual tokens. Meanwhile, the transformer decoder and diffusion model contribute to high diversity through synonym replacement and sentence restructuring. The analysis has been included in Lines 481-506 of the revised paper.
>
> **[Q5] The writing is somewhat verbose. For instance, in Subsection 3.6, the second sentence ...**
> We appreciate your feedback regarding the verbosity of the writing. We have improved the sentences in Subsection 3.6 of the revised paper and will work to improve the overall conciseness of the paper.
>
> **[Q6] Some equations could be improved; for example, Equations 9 and 10 differ by only one symbol.**
> Thank you for pointing out this issue. In the revised paper, Eq. 9 is kept, while Eq. 10 is replaced by simply stating its difference from Eq. 9 in Line 294.
>
> **[Q7] There are a few typos, such as a missing period on line 82 and an incorrect number on line 360.**
> Thank you. We have corrected the typos in the revised paper and will proofread the entire paper.

---

> > ### Author Response · Authors · 2024-11-25
> > **We hope to have your further feedback on our replies**
> >
> > We have made every effort to address your concerns and sincerely hope our responses meet your expectations. We would greatly appreciate your feedback on our replies to help us further enhance the manuscript.

---

> > > ### Author Response · Authors · 2024-11-29
> > >
> > > Dear Reviewer,
> > >
> > > We have provided detailed responses to your comments and hope they address your concerns. We look forward to receiving your further feedback on our replies.

---

### Meta-Review · Area_Chair_ioW7 · 2024-12-22

**Metareview:**

Paper explores the integration of referring expression generation and comprehension tasks. The paper was reviewed by four expert reviewers and received: 3 x  marginally above the acceptance threshold and 1 x accept, good paper ratings. Reviewers agree that the approach is sound and experiments through. Most of the reviewer comments centered around: (1) overall novelty of the approach, (2) analysis of results and (3) lacking comparison to SoTA (e.g., CCL [Wang et al. AAAI'24]). Authors have provided a through rebuttal with reviewers generally satisfied with the responses. For the most part, only (1) remains a concern for [5XtB].

AC has read the reviews, rebuttal and discussion that followed; and also looked at the paper itself. Overall, AC agrees with the positive consensus of reviewers and believes that the use of diffusion for the task is innovative (even if individual components may not necessarily be so). As a results, AC is recommending Acceptance.

Authors are encouraged to incorporate results from the rebuttal and discussion into the main paper.

**Additional Comments On Reviewer Discussion:**

Authors have provided a through rebuttal with reviewers generally satisfied with the responses. Specifically, reviewer [jswi] states that his/her "concern has been addressed" and the "positive score" will be maintained. Reviewer [5XtB] mentions that "the effectiveness of the proposed method has been clearly demonstrated", but also notes that "there are still no major changes to the points raised in Q1". As a result, [5XtB] updated the score from 5 to 6. Reviewer [yjk1] was also positive and [XHYa], while did not respond, did raise the rating post-rebuttal. Overall, the sentiment post-rebuttal from the reviewers was positive, which has ultimately led to recommendation above.

---

### Decision · Program_Chairs · 2025-01-22

Accept (Poster)